# Complete Range of the Universal mtDNA Gene Pool and High Genetic Diversity in the Thai Dog Population

**DOI:** 10.3390/genes11030253

**Published:** 2020-02-27

**Authors:** Liangliang Zhang, Yilin Liu, Quan Thai Ke, Arman Ardalan, Ukadej Boonyaprakob, Peter Savolainen

**Affiliations:** 1Department of Gene Technology, KTH–Royal Institute of Technology, Science for Life Laboratory, 171 21 Solna, Sweden; liangzhang588@gmail.com (L.Z.); yilin.liu@scilifelab.se (Y.L.); ardalana@kth.se (A.A.); 2Department of Natural science Education, SaiGon University, Ho Chi Minh city 999100, Vietnam; quan.tk@cb.sgu.edu.vn; 3Department of Physiology, Faculty of Veterinary Medicine, Kasetsart University, Bangkok 10700, Thailand; fvetudb@ku.ac.th

**Keywords:** dog, mtDNA, Thailand, genetic diversity

## Abstract

The dog population of Southern East Asia is unique in harboring virtually the full range of the universal mtDNA gene pool, and consequently, it has the highest genetic diversity worldwide. Despite this, limited research has been performed on dog genetics within this region. Here we present the first comprehensive study of a sub-region within Southern East Asia, analyzing 528 bp of mtDNA for 265 dogs from Thailand, in the context of dogs from across the Old World. We found that Thailand was the only region in the world that has the full range of the universal mtDNA gene pool, that is, all 10 sub-haplogroups. Consequently, the statistics for diversity are among the highest, especially in North Thailand, which had high values for haplotype diversity and the number of haplotypes, and the lowest proportion of individuals with a universal type-derived haplotype (UTd) among all regions. As previously observed, genetic diversity is distinctly lower outside Southern East Asia and it decreases following a cline to the lowest values in western Eurasia. Thus, the limited geographical region of Thailand harbors a distinctly higher genetic diversity than much larger regions in western Eurasia, for example, Southwest Asia and Europe which have only five and four of the 10 sub-haplogroups, respectively. Within Thailand, diversity statistics for all four sub-regions follow the general pattern of Southern East Asia, but North Thailand stands out with its high diversity compared to the other regions. These results show that a small part of Southern East Asia harbors the full range of the mtDNA gene pool, and they emphasize the exceptional genetic status of Southern East Asia. This indicates that today’s dogs can trace a major part of their ancestry to Southern East Asia or closely situated regions, highlighting Thailand as a region of special interest. Considering the large genetic diversity found in Thailand and that many neighboring regions, e.g., Myanmar and Laos, have not been studied for dog genetics, it is possible that large parts of the dog gene pool remain undiscovered. It will be an important task for future studies to fill in these blanks on the phylogeographic map.

## 1. Introduction

The domestic dog is considered to be the first domesticated animal and was the only domesticate to accompany humans to every continent in ancient times. Despite this central role in human history, and considerable efforts in previous research, the question about where the dog originated remains unresolved. Previous global studies of mitochondrial DNA (mtDNA) variations among dogs have suggested they originated in East Asia [1], and more specifically, the region south of the Yangtze River [2], which spans the southern part of China and mainland Southeast Asia, a region that we call Southern East Asia. This was based on the unique presence in this region of the full phylogenetic range of diversity, all 10 sub-haplogroups (a1–a6, b1, b2, c1 and c2) of the three globally distributed haplogroups A, B and C [2]. mtDNA is a single genetic marker only inherited through the female line. Consequently, additional datasets need to be analyzed using independent markers to verify the mtDNA-based results. However, a study of global Y-chromosome diversity showed the highest genetic diversity in Southern East Asia [3] and a more recent study of the nuclear genome sequence in 58 dogs and wolves showed that dogs from Southern East Asia had the basal phylogenetic position relating to wolves and the highest genetic diversity [4], thus giving three independent datasets that suggest Southern East Asia as a possible place of origin for dogs. On the other hand, several other studies have claimed other geographical origins of dogs: a study of genome wide SNPs suggested the Middle East [5], a study of mtDNA from archaeological samples suggested Europe [6] and a study of genome wide SNPs suggested a region encompassing Nepal and Mongolia [7]. However, none of these studies included comprehensive samples from Southern East Asia.

Thus, the geographical origin of the domestic dog is under vigorous debate, but the mitochondrial data is clear in that Southern East Asia is unique in harboring virtually the full range of the universal mtDNA gene pool, and consequently, it has the highest genetic diversity worldwide for this marker. Therefore, in order to obtain a comprehensive picture of the global gene pool for mtDNA, studies of Southern East Asia and the surrounding regions are vital. Despite this, research on dog genetics in this area is limited so far.

Thailand is located in the center of mainland Southeast Asia, in the southern part of Southern East Asia. It has four distinctive regions, each with unique geographic and cultural characteristics: North, Northeast, Central and South Thailand. North Thailand is the most isolated and desolate part of the country with hilly forested areas and a tropical savanna climate. Northeast Thailand is dominated by a vast and relatively arid plateau. Central Thailand is dominated by a large fertile plain. South Thailand comprises a substantial portion of the Malay Peninsula and Thailand’s coastline with major tourist industry [8]. Thailand’s potential role as a corridor between southern China and Island Southeast Asia is indicated by archaeological evidence for agricultural communities that may have expanded from the center of the Yangtze valley during the Neolithic period [9]. Native dogs in the rural parts of Thailand are usually free-breeding, but have specific owners and are normally used as guard dogs. They have a relatively uniform exterior, with a slim body shape, long narrow legs, pointed ears and tails, short hair, and an alert and watchful temperament. The Thai ridgeback dog is an old breed of dog from Thailand, named for its characteristic dorsal hair ridge. The origin of this breed, and its ridge genotype, is of great cultural, historical and genetic interest but is still unknown. Globally, there are two additional populations with similar hair ridge, the Rhodesian ridgeback and the Vietnamese Phu Quoc dog. The ridge mutation has been shown to be identical by descent in all three breeds, but its origin is unknown [10].

Considering the exceptional genetic diversity of the mtDNA pool of Southern East Asia and the relatively sparse sampling of this large region, which is half the size of Europe, it is clear that expanded studies of Southern East Asia are a prerequisite for obtaining a full picture of global mtDNA diversity. Previously, no comprehensive genetic studies of village dogs in Thailand or any other part of Southern East Asia, or of the Thai ridgeback dog, have been reported. We therefore performed the first large-scale study of a sub-region within Southern East Asia, analyzing 528 bp of mtDNA for 265 dogs from Thailand, in the context of dogs from across the Old World. Hereby, we give the first more detailed description of the mtDNA gene pool in a sub-region of Southern East Asia, thus shedding light on the genetics in a region of central importance for studies of the origins of today’s domestic dogs, and contributing to a comprehensive picture of the global mtDNA gene pool.

## 2. Materials and Methods 

### 2.1. Samples

A total of 265 samples for dogs from Thailand (163 new samples and 102 samples from the literature) were analyzed for a 582bp fragment of the control region of the mitochondrial genome (positions 15,458–16,039). This region of mtDNA is the only genetic marker studied so far for a comprehensive collection of dogs from across the world. Therefore, we were able to study the Thai dogs in the context of samples from across the Old World published in previous studies [2,11,12,13], in total 3254 dogs and 40 wolves. As shown in Figure 1, the Thai samples were collected from across large parts of Thailand. We divided the samples into four distinctive geographical sub-regions, each with unique geographic and cultural characteristics: North (*n* = 80), Northeast (*n* = 60), Central (*n* = 32) and South (*n* = 61). We also included samples from one indigenous dog breed, the Thai ridgeback (*n* = 28). A list of all samples with information about, e.g., geographical region and mtDNA haplotype is given in Appendix A. The sampled dogs were, except for a few exceptions, collected from remote rural regions with a limited influx of foreign dogs. Therefore, the sample collection from Thailand offers, to our knowledge, a good representation of the indigenous Thai dog population.

### 2.2. DNA Extraction, Amplification and Sequencing

Samples were collected as buccal cell samples on FTA cards (Whatman Inc., Maidstone, UK) according to the manufacturer’s instructions. The target sequence was amplified using a nested PCR reaction suggested by [12]. The outer primers for amplification are H15404 (5′-CCT AAG ACT CAA GGA AGA AGC-3′) and L16102 (5′-AAC TAT ATG TCC TGA AAC CAT TG-3′), and primers H15430 (5′-TCC ACC ATC AGC ACC CAA AG3′) and L16092 (5′-CTG AAA CCA TTG ACT GAA TAG-3′) for the inner. Two pairs of primers were used for sequencing, an inner primer from the nested PCR H15430 and L16092, and two additional primers: H15706 (5′-CAC CAT GCC TCG AGA AAC CAT-3′) and primer L15791 (5′-ATG GCC CTG AAG TAA GAA CC-3′). Sequencing was performed using ABI Big Dye terminator chemistry and analysis on the ABI 3700 DNA sequencer as described by [12].

### 2.3. Phylogenetic and Statistical Analysis

The DNA sequences were manually aligned using Mega X. A neighbor-joining (NJ) tree was constructed by Mega X [14], using Coyote [2] as the outgroup. Calling of haplotypes was done using DnaSP 6.12.1 [15]. The novel haplotypes found in this study were deposited in GenBank under accession numbers MN603428–MN603444 (haplotypes A228–A241, B050, B051, and E005, respectively). The minimum spanning (MS) network, showing the shortest genetic distance (in the number of substitutions) between haplotypes, was calculated by Arlequin and drawn manually. An additional MS network, including samples from Nigeria [11], was made with PopArt [16] (Appendix A). Haplotype diversity, nucleotide diversity and pairwise genetic distances (with SD) were estimated using Arlequin 3.5 [17]. To compare the number of haplotypes among populations with different sample sizes, sample sizes were adjusted by resampling without replacement (1000 replications) using an in-house developed program. 

## 3. Results

### 3.1. The Thai Dog Population Belongs to the Universal mtDNA Gene Pool of the Old World

We analyzed 582bp of the mtDNA control region for 265 dogs from Thailand in the context of samples from across the Old World, that is, a total of 3254 dogs. As described in previous studies, the mtDNA gene pool was, at a superficial level, remarkably homogenous among geographical regions. Phylogenetic analysis showed three principal phylogenetic groups, haplogroup A, B, and C (Figure 2), which are represented in all populations across the Old World in 98% of the dogs (Table 1). Furthermore, these three haplogroups are represented at similar frequencies across populations, normally in the ranges of 60–80% for haplogroup A, 10–30% for B, and 5–15% for C (Table 1). The remaining 2% of the dogs had haplotypes belonging to the three rare and geographically restricted phylogenetic groups D, E and F. It is also noteworthy that a majority of all dogs (62.2%) carried one of just 18 haplotypes (out of total of 369 haplotypes), which are almost universally occurring (they are shared by the European, Southwest Asian and East Asian populations and have therefore been denoted “universal type”, UT [1] (Figure 3, Table 1). Furthermore, 77% of all dogs, and over 90% in some regions, e.g., Europe and Southwest Asia, carried a haplotype, which is either a UT or differs by a single mutation from one of the UTs (collectively denoted as a "UT-derived haplotype", UTd) (Table 1). In summary, the mtDNA gene pool was homogenous across the Old World, with 98% of all dogs sharing the three principal haplogroups at similar proportions, and 77% carrying a haplotype that can be traced to just 18 universally occurring UT-haplotypes. This implies that today’s domestic dogs share a common genetic origin from a universally shared gene pool. 

The Thai dog population fits well in this universal phylogeographic pattern. Of the 265 dogs from Thailand, 163 were new samples, among which 17 new haplotypes were identified: 14 in haplogroup A, 2 in B, and 1 in E. In total, 65 haplotypes were found, which were assigned to haplogroups A, B, C, and E, and these haplogroups were represented by 83.0%, 9.8%, 4.9%, and 2.3% of the dogs, respectively. Among these 65 haplotypes, 13 were UTs and 32 were unique to Thailand (Figure 2, Figure 3 and Table 1). 

Thus, the Thai dog population shares the universal mtDNA gene pool, and follows the general pattern in that almost 100% of the dogs carry haplotypes belonging to haplogroups A, B and C and these three haplogroups are represented at proportions similar to most other regions.

### 3.2. The Full Range of the Genetic Diversity for the Universal mtDNA Gene Pool was Found Only in Thailand

However, as it has already been reported, while the dog population of the Old World was overall strikingly homogenous at a superficial phylogenetic level in sharing the same three principal haplogroups at similar proportions, there were also very distinct differences at a more detailed phylogenetic level. Phylogenetic studies of mitochondrial genomes have shown that the three haplogroups A, B and C have a more detailed phylogenetic substructure. This divides these three haplogroups into a total of 10 distinct sub-haplogroups (a1–a6, b1, b2, c1 and c2), which can also be distinguished for the analyzed 582 bp region based on diagnostic mutations [2], and in the minimum spanning network (Figure 3). Notably, in the phylogenetic tree sub-haplogroups a2, a3, a4, a6 and c2 form separate clades but with low bootstrap values (Figure 2).

In contrast to haplogroups A, B and C, these 10 sub-haplogroups were not universally distributed in the Old World, instead, the distribution followed a distinct cline with a maximum in Thailand. Thus, the only region harboring all 10 sub-haplogroups was Southern East Asia, and within Southern East Asia, the Thai dog population stood out as the only subpopulation that harbored all 10 sub-haplogroups (Figure 3, Figure 4, Figure 5a and Table 1). From the "epicenter" of Thailand, the distribution followed an approximate gradient in all directions with, e.g., 9 sub-haplogroups in South China, 7 in Central China and Island Southeast Asia, 6 in Japan and India, down to only 5 sub-haplogroups in Southwest Asia and 4 in Europe (Figure 4). For the largest haplogroup, haplogroup A, the difference in the distribution of sub-haplogroups was especially accentuated in that most regions, e.g., SW Asia and Europe, harbored only one of the six sub-haplogroups, sub-haplogroup a1 (Figure 3). It is also noteworthy that the representation of haplotypes and sub-haplogroups are very similar for Europe and SW Asia (Figure 3). Therefore, the low genetic diversity for European dogs does not seem to be caused by biased sampling of breed dogs, but must stem from a time before the formation of the European and SW Asian dog populations from a common source.

The cline for the number of sub-haplogroups was reflected by other statistics of genetic diversity: the proportion of UTs and UTds, the number of haplotypes (adjusted for sample size by resampling, e.g., nHTres59) and haplotype diversity (Figure 5, Table 1, Appendix A). The proportions of UT and UTd differed dramatically among different regions, with very high proportions, mostly above 90% UTd, in the western part of the Old World, e.g., Europe and Southwest Asia, following a cline to around 80% in Central Asia and East Asia to below 40% in Southern East Asia, with a minimum of 33.8% in North Thailand. This implies that almost all haplotypes in the western part of the Old World can be traced to an origin from just 18 haplotypes (the 18 UTs) that are also found in East Asia while in Southern East Asia the majority of haplotypes are genetically distinct from those in the West.

Also with regard to the number of haplotypes and the haplotype diversity, Southern East Asia stood out with much higher values than other regions, except for Siberia, which had the highest haplotype diversity. Within Southern East Asia, Thailand stood out also for these statistics, with higher values than South China. The only major deviation from the general trend in the data was the Nigerian population, which was obtained from a recent article by Adeola et al. [11]. Our study includes a comprehensive sample of 528 dogs from other regions across sub-Saharan Africa, which had values following quite closely the clines for the diversity measures. However, the Nigerian samples had numerous unique haplotypes, a larger number of haplotypes (nHTres59) and a lower proportion of UTs and UTds than expected from the general trend, but harbored only three of the ten sub-haplogroups. Thus, they differed distinctly from the samples from other parts of sub-Saharan Africa (Appendix A, Figure 5a, Table 1).

### 3.3. Sub-Populations of Thailand and Southern East Asia

Within Thailand, the statistics for all four sub-regions conform largely to the values generally found in Southern East Asia, e.g., concerning the proportion of individuals carrying haplogroups A, B and C, and the proportions carrying UTs and UTds, but there were also some clear differences in the genetic diversity among the four sub-regions. Most importantly, North Thailand had a higher diversity compared to the other regions, especially compared to Northeast and South Thailand, with regard to the number of haplotypes and haplotype diversity as well as the percentage of dogs carrying UTs and UTds. Especially the low proportion of dogs carrying a UT or UTd stands out, with the lowest numbers of all populations in the study. The lowest genetic diversity was found in Northeast Thailand, which had a low number of haplotypes and only 5 of the 10 sub-haplogroups, compared to 7–8 in the other regions.

We also compared the sub-regions of Thailand with other parts of Southern East Asia, specifically three parts of South China: Yunnan, Guizhou and Guangxi, and Hunan and Jiangxi. These regions all had a high number of sub-haplogroups, that is, 9, 8 and 8, respectively. For the other statistics, North Thailand stood out with the highest haplotype diversity and lowest proportion of individuals carrying UTs and UTds, and second highest number of haplotypes. Notably, although Yunnan had 9 sub-haplogroups, the highest number of all sub-regions, it had the lowest haplotype diversity, possibly a result of biased sampling from related individuals (Figure 5b, Table 1). Overall, the diversity was relatively homogenous across Southern East Asia, with six of the seven regions having 7–9 sub-haplogroups, and all regions having 60% or less of individuals carrying UTds. However, it is notable that North Thailand and Yunnan, which had the highest haplotype diversity and lowest proportion of individuals with a UT or UTd, and the highest number of sub-haplogroups, respectively, are both situated in the northwestern part of Southern East Asia.

The Thai ridgeback dogs had a similar proportion of haplogroups A and B as the non-breed dogs in Thailand, with 85.7% haplogroup A, 14.3% haplogroup B, but lacked clade C. They also had a similar number of dogs carrying UTs and UTds. With 13 haplotypes among the 28 Thai ridgeback dogs, the genetic diversity was significantly higher than for European dog breeds, which normally have around 5 haplotypes [18].

### 3.4. Haplogroup E Diversity is Centered in Southeast Asia

Haplogroup E was rare, both in the total sample (a total of 16 dogs out of the 3254 included in this study; 0.49%) and in the Thai sample (6 dogs out of 265; 2.3%), and was found exclusively in East Asia: in Thailand, Vietnam, Indonesia, Korea and Japan. All four E haplotypes were found in mainland and island Southeast Asia, and three of the four were exclusively in this region; only the most frequent haplotype, E1 was found in Korea and Japan. Furthermore, three of the four haplotypes were found in Thailand while only a single one was found in any other region. It is also noteworthy that the E haplotypes were absent in the sample of 282 dogs from South China.

With only 16 individuals carrying E haplotypes no certain conclusions can be drawn about the phylogeography of haplogroup E. However, the fact that all four haplotypes were found in mainland and island Southeast Asia but only a single haplotype was found outside this region, and that the haplogroup was absent in South China, indicates that haplogroup E originated from wolves somewhere in mainland Southeast Asia.

## 4. Discussion

This study demonstrates that the relatively small geographical region of Thailand, which is only 5% the size of Europe, uniquely harbors virtually the whole range of the universal mtDNA gene pool, that is, all 10 sub-haplogroups. This reinforces the distinct difference in the distribution of mtDNA haplotypes across the Old World, that populations in Southern East Asia have a much more complex diversity than other regions. It also, for the first time, specifically highlights the sub-regions south of China as areas of special interest. 

Evidence from whole nuclear-genome data has shown that dogs from Southern East Asia have the basal phylogenetic position relating to wolves and the highest genetic diversity [4], and that dogs from Nepal and Mongolia have the lowest short-range linkage disequilibrium [7]. Together with the mtDNA data [3], this suggests that the gene pool of the modern dog population may have dispersed from southern or eastern Asia, and also suggests the possibility that this region could be the source of the ancient gene pool and the origin of the domestic dog.

While Thailand is unique in harboring all 10 sub-haplogroups, two of these, a4 and a6, are rare, and the absence of one of these in the sample from South China may reflect incomplete sampling. Thus, the difference among the sub-regions within Southern East Asia was not dramatic. However, it is notable that North Thailand and Yunnan, which are both situated in the northwestern part of Southern East Asia, had the highest haplotype diversity and lowest proportion of individuals with a UT or UTd, and the highest number of sub-haplogroups, respectively. This suggests the northwestern part of Southern East Asia as a possible source for the modern mtDNA gene pool. However, it also highlights the total lack of samples from large parts of Southern East Asia. Two specific examples are the neighboring regions, Myanmar, which borders the entire western side of Yunnan and the northwestern part of North Thailand, and northern Laos, situated between southern Yunnan and North Thailand. Considering the unique diversity in North Thailand and Yunnan, it is possible that a considerable part of the dog mtDNA gene pool remains undiscovered in Myanmar and Laos, and future studies will need to fill in these and other blanks on the phylogeographic map of southern and eastern Asia.

Extended sampling is also warranted in South China. Only 282 samples from this vast region have been analyzed so far, and detailed sampling information is largely lacking detail, indicating nothing more than the province [2]. Thus, it is possible that samples were collected from just a few areas within each province, implying that samples may come from related individuals, which may explain the extremely low haplotype diversity for the sample from Yunnan.

A neighboring region of special interest is Nepal and Mongolia, referred to as Central Asia in [7], a study that suggested that dogs probably originated from this region based on SNP analysis of nuclear DNA showing the lowest short-range linkage disequilibrium in this region. The study included analysis of mtDNA, but based on SNPs rather than complete DNA sequences, and therefore the results are not directly comparable to most of the statistics in our study. However, the number of sub-haplogroups can be investigated based on diagnostic mutations. Thus, we combined the 26 samples from [7] with 7 samples from our study, giving a total of 33 samples from Nepal and Mongolia. Six of the 10 sub-haplogroups were found among these samples, following the general phylogeographic trend. However, the sample size is small and it is possible that sub-haplogroups remain undetected because of incomplete sampling. 

This study shows that, except for the 2% of the dogs that carry haplotypes belonging to haplogroups D, E and F, there are no region-specific haplogroups, which implies that dogs worldwide share a universal gene pool. This in turn implies that all dogs in today’s global domestic dog population originate from the same domesticated wolves, which were either domesticated in a single region, or in different regions for the different haplogroups followed by sufficient mixing of haplotypes. In conclusion, there is clearly a single genetic origin for today’s dog population, though not necessarily from a single geographical region.

It is important to appreciate that this is a study of today’s dog population. The fact that practically the full range of today’s genetic diversity for mtDNA is found in Southern East Asia is a strong indication that the modern mtDNA gene pool originates from there, or from neighboring regions. However, this does not necessarily mean that the original mtDNA gene pool, and thus the domestic dog, also actually originated there. It is possible that the universal mtDNA gene pool experienced considerable changes during the ancient history of dogs. An indication of this is the dramatic change through time in the proportions of haplogroups A, B and C in Europe. A study of mtDNA in European pre-Neolithic dogs found all 15 investigated samples (100%) to have haplotypes belonging to haplogroup C [19]. In contrast, a study of 16 Neolithic dogs from Southeast Europe showed “normal” frequencies of haplogroup A and B (75% and 20%, respectively) and no haplotype from haplogroup C [20]. Our study similarly shows that today’s European dogs carry all three major haplogroups, A, B and C, with only 8% carrying haplotype C (Table 1).

Nevertheless, the fact that the full range of sub-haplogroups of haplogroups A, B and C have been found exclusively in Southern East Asia indicates this region, or neighboring regions, as a probable origin for much of the modern dog mtDNA gene pool. Together with the evidence from Y-chromosome and whole nuclear-genome data [3,4,7], this also suggests that Southern East Asia with its neighboring regions is a possible candidate to be the actual origin of dogs. Exactly where in southern or eastern Asia this would be is not clear, because there are still large blanks on the map that totally lack phylogenetic studies of dogs, e.g., Myanmar and Laos, and several regions with very limited sampling, e.g., the Himalayan region with Nepal, northern India and Tibet. However, the exclusive presence of all 10 sub-haplogroups in Thailand specifically includes Thailand within this candidate region. Whether it is a plausible theory that the dogs may have originated from Thailand depends, among other things, on whether or not wolves were ever present in Thailand. Since the dog originated from the wolf, the historical range of the wolf puts an obvious limit to the possible regions where dogs may have originated. 

The southern range of wolves in Southern East Asia is unclear. Western literature has incorrectly stated that wolves are absent from today´s Southern East Asia, with some articles claiming that wolves were never present in this region, even in ancient time. However, this was recently corrected in a review of the distribution of the wolf in China [21], showing that the southernmost record of wolf in modern time is in Yunnan in South China, only 200 km north of Thailand. This suggests a theoretical possibility that the domestic dog may have originated as far south as Thailand. However, no remains of wolf have been found in the sparse archaeological records from northern Thailand [9]. An alternative scenario, which is perhaps more probable, is that the dog originated slightly more to the north, but close enough to preserve most of the genetic diversity after migration into Thailand. This scenario correlates with the history of the Thai people, which probably began with the Tai people migrating from southern China [8]. It is clear that many parameters are unknown with regard to the place and time of the dog’s origins, and that an open mind must be kept until even the unlikely scenarios are definitely dismissed.

Consequently, this study identifies two important tasks in the quest for the genetic origins of the modern as well as the ancient dog population: to describe the historical spread of the wolf in southern and eastern Asia, and to fill in large blanks on the phylogeographical map of dogs in this region. In addition, the exceptional mtDNA diversity, compared to other sub-Saharan populations, of the Nigerian dog population warrants detailed studies of mitochondrial genomes in sub-Saharan Africa.

In conclusion, this study demonstrates that the relatively small geographical region of Thailand is the only region in the world that has been shown to harbor the full range of the universal dog mtDNA gene pool, all 10 sub-haplogroups. This reinforces the exceptional genetic status of the dog population in Southern East Asia, which has already been indicated in previous studies of mtDNA, Y-chromosome and whole nuclear-genome data, and highlights Thailand as a possible region from which today’s global dog population may have dispersed. However, large parts of southern and eastern Asia remain poorly studied, or are even blind spots on the phylogeographical map. Therefore, to get a comprehensive picture of the global gene pool of today’s domestic dogs, further studies in and around Southern East Asia, of mitochondrial as well as nuclear genomes, will be essential.

## Figures and Tables

**Figure 1 genes-11-00253-f001:**
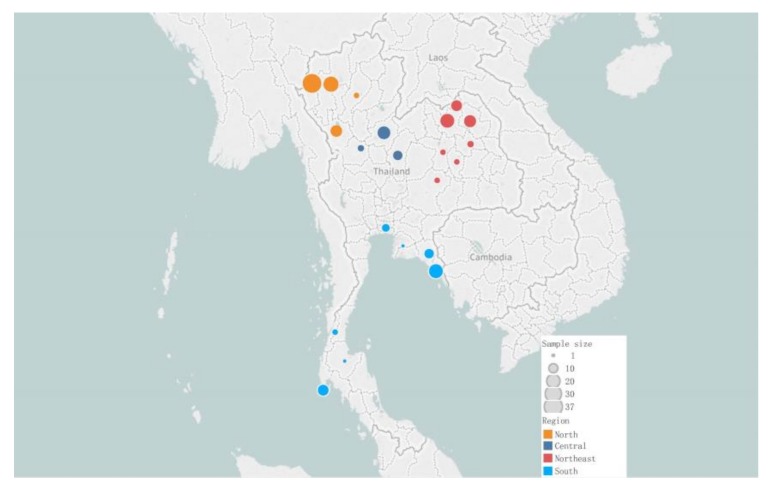
Map of Thailand showing sampling location and sample numbers.

**Figure 2 genes-11-00253-f002:**
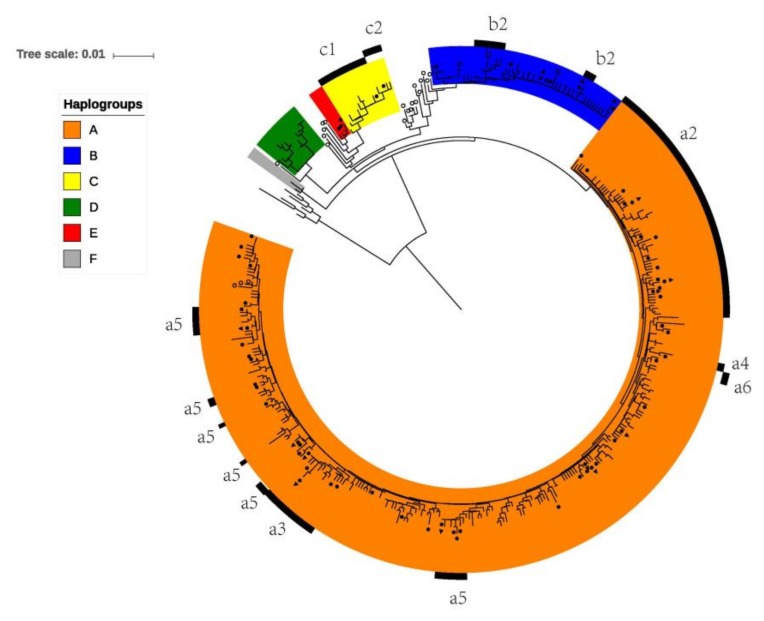
A neighbor-joining (NJ) tree for the dog and wolf haplotypes, rooted by coyote sequences. The six dog haplogroups are indicated by colors. Sub-haplogroups (according to diagnostic mutations) are indicated by black lines and the respective name, except a1 and b1 which consists of the unmarked parts of haplogroup A and B, respectively. Black dots indicate haplotypes found in Thailand, triangles indicate haplotypes found in Thai ridgeback dogs, and white circles indicate haplotypes found in wolves. Bootstrap values were 62%, 70%, 66% for haplogroups C, D and F, respectively, and 63% and 57% for sub-haplogroups c2 and a4, respectively, and below 50% for all other haplogroups and sub-haplogroups.

**Figure 3 genes-11-00253-f003:**
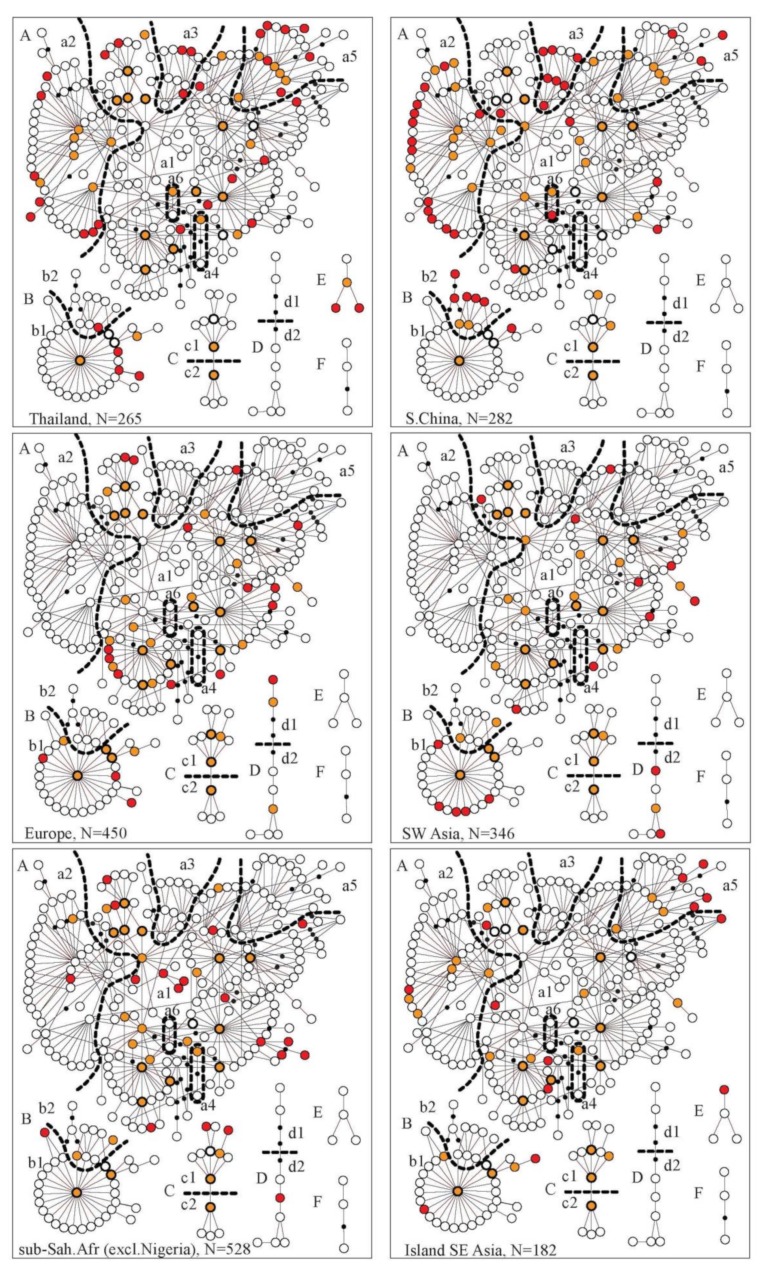
Minimum spanning (MS) networks showing genetic relationships among haplotypes and their geographical distribution. Circles represent haplotypes, lines connecting haplotypes represent a single substitution step, and black dots represent hypothetical intermediary haplotypes. The 18 universal haplotypes (UTs) are indicated by black bold circles. The six haplogroups A–F are presented as separate networks. The ten sub-haplogroups, a1–a6, b1 & b2, c1 & c2 are separated by dashed lines. For each geographical region, colored circles indicate haplotypes represented in that region, orange indicates haplotypes shared with other regions and red indicates haplotypes unique to the region. For clarity of the MS network, haplotypes unique to Nigeria are excluded (see an MS network including all haplotypes in Appendix A).

**Figure 4 genes-11-00253-f004:**
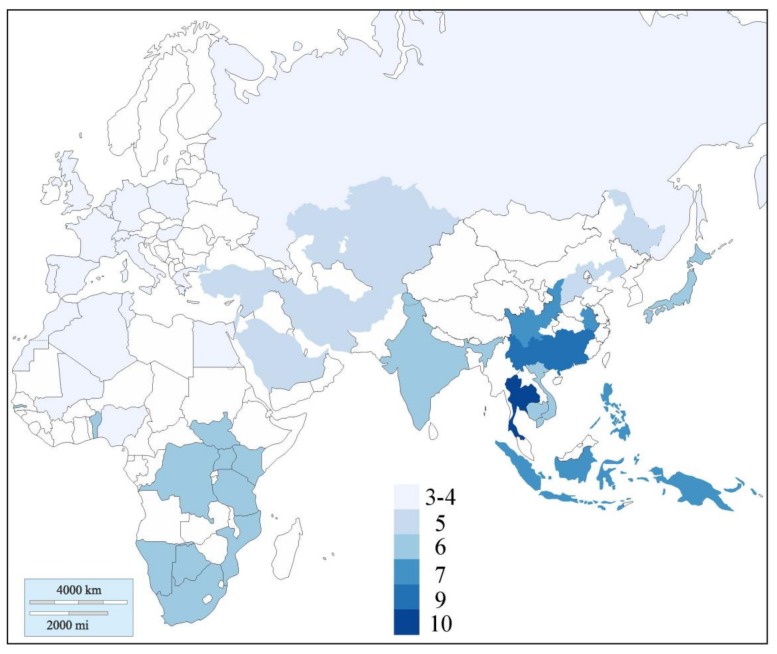
World map showing the number of sub-haplogroups found in different regions.

**Figure 5 genes-11-00253-f005:**
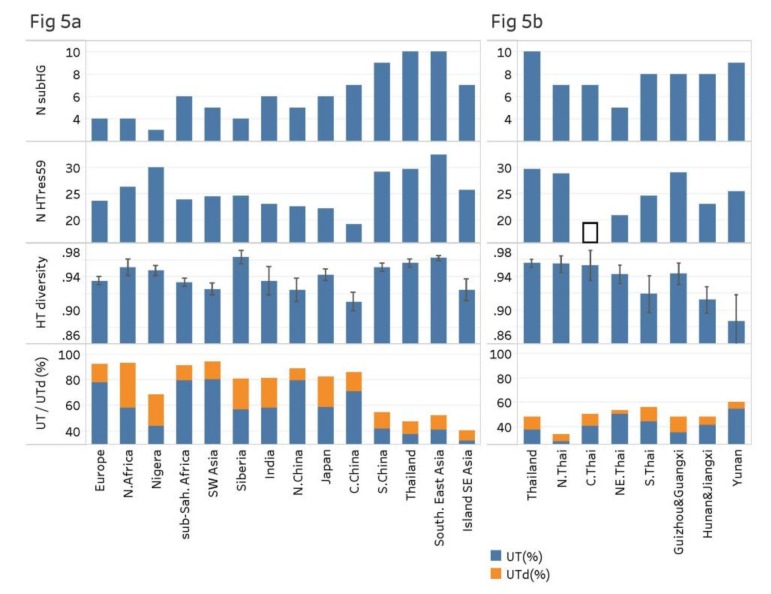
Genetic diversity in geographical regions: number of sub-haplogroups, number of haplotypes from resampling with sample size 59, Haplotype diversity and proportion of individuals carrying a UT and Utd. (**a**) Regions across Eurasia and Africa. (**b**) Sub-regions within Southern East Asia. Black square indicates the number of haplotypes found in the 32 samples from Central Thailand.

**Table 1 genes-11-00253-t001:** Diversity statistics for the investigated regions.

Region	ABC (DEF) ^1^	nA (%) ^2^	nB (%) ^2^	nC (%) ^2^	nHT ^3^	HTuq ^4^	UT (%) ^5^	UTd(%) ^5^	NsubHG ^6^	nHTres59 ^7^	HT Diversity (SD) ^8^
Total	3185 (69)	2412 (74.1)	547 (16.8)	226 (6.9)	369	-	62.2	77.0	-	-	-
Europe	423 (27)	292 (64.9)	95 (21.1)	36 (8.0)	62	16	77.8	92.2	4	23.60	0.935 (0.005)
N.Afr^9^	83 (3)	60 (69.8)	12 (14.0)	11 (12.8)	31	10	58.1	93.0	4	26.27	0.951 (0.010)
Nigeria	336 (9)	319 (92.5)	15 (4.3)	2 (0.6)	75	54	43.8	68.4	3	30.04	0.947 (0.006)
Sub-Sah. Africa^10^	527 (1)	396 (75)	98 (18.6)	33 (6.3)	61	21	79.2	91.1	6	23.95	0.933 (0.005)
SW Asia^11^	337 (9)	200 (57.8)	115 (33.2)	22 (6.4)	59	15	79.8	94.2	5	24.48	0.925 (0.007)
Siberia	60 (2)	39 (62.9)	13 (21,0)	8 (12.9)	25	7	56.5	80.7	4	24.62	0.963 (0.008)
India	59 (0)	47 (79.7)	4 (6.8)	8 (13.6)	23	6	57.6	81.4	6	23	0.935 (0.017)
Island SEA^12^	181 (1)	138 (75.8)	22 (12.1)	21 (11.5)	46	13	32.4	40.7	7	25.77	0.924 (0.013)
Thailand	259 (6)	220 (83.0)	26 (9.8)	13 (4.9)	67	32	37.4	47.5	10	29.71	0.956 (0.005)
C.Thai^13^	31 (1)	25 (78.1)	3 (9.4)	3 (9.4)	17	6	40.6	50.0	7	-	0.954 (0.018)
N.Thai^14^	79 (1)	67 (83.8)	8 (10.0)	4 (5.0)	35	15	27.5	33.8	7	28.95	0.955 (0.010)
NE.Thai^15^	60 (0)	49 (81.7)	9 (15.0)	2 (3.3)	21	5	50.0	53.3	5	20.87	0.943 (0.011)
S.Thai^16^	58 (3)	52 (85.2)	2 (3.3)	4 (6.6)	25	5	44.3	55.7	8	24.58	0.919 (0.022)
Thai ridgeback	28 (0)	24 (85.7)	4 (14.3)	-	13	2	17.9	53.6	5	-	0.931 (0.024)
Japan	118 (3)	76 (62.8)	24 (19.8)	18 (14.9)	30	7	58.7	82.6	6	22.27	0.942 (0.007)
N.China	98 (0)	65 (66.3)	25 (25.5)	8 (8.2)	29	4	79.6	88.8	5	22.64	0.924 (0.014)
C.China	141 (0)	109 (77.3)	21 (14.9)	11(7.8)	29	8	70.9	85.8	7	19.27	0.910 (0.011)
S.China	282 (0)	223 (79.1)	45 (16.0)	14 (5.0)	74	35	41.8	54.3	9	29.24	0.951 (0.005)
Guizhou/Guangxi	92 (0)	76 (82.6)	13 (14.1)	3 (3.3)	38	16	34.8	47.8	8	29.02	0.944 (0.013)
Hunan/Jiangxi	100 (0)	68 (68.0)	26 (26.0)	6 (6.0)	31	9	41.0	48.0	8	23.03	0.912 (0.016)
Yunan	75 (0)	68 (90.7)	4 (5.3)	3 (4.0)	29	8	54.7	60.0	9	25.53	0.887 (0.032)
Southern East Asia	580 (7)	469 (79.9)	78 (13.3)	33 (5.6)	127	73	40.7	52.0	10	32.39	0.962 (0.003)

^1^ Total number of samples divided for the universal haplogroups A, B, and C (non-universal haplogroups D, E, and F). ^2^ Number of samples belonging to each of haplogroups A, B, and C (proportions of the total samples of the region). ^3^ Number of haplotypes. ^4^ Number of unique haplotypes. ^5^ Proportion of individuals carrying a universal type (UT)/UT-derived haplotype (UTd). ^6^ Number of sub-haplogroups. ^7^ Number of haplotypes obtained from resampling of size 56 (the size of the smallest sample) with 1000 replications. ^8^ Haplotype diversity (standard deviation). ^9^ North Africa. ^10^ Sub-Saharan Africa. ^11^ Southwest Asia. ^12^ Island Southeast Asia. ^13^ Central Thailand. ^14^ North Thailand. ^15^ Northeast Thailand.^16^ South Thailand.

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
