# Peer review of "Complete Range of the Universal mtDNA Gene Pool and High Genetic Diversity in the Thai Dog Population"

_genes, 2020, doi:10.3390/genes11030253_

Round 1

Reviewer 1 Report

The article presents a new mtDNA data about recent Thailand villages dogs, n=163 new and one native dog breed (Thai ridgeback dog, n=28). There was identified all 10 dog mtDNA sub-haplogroups (second level) in Thai dog’s gene pool and high genetic diversity. The authors discussed this interesting fact in contexts to one of hypothesis for the single point origin/domestication of the dogs in South-East Asia and migration worldwide.    

In conclusion, this study demonstrates that the relatively small geographical region of Thailand is the only region in the world so far shown to harbor the full range of the universal dog mtDNA gene pool, all 10 sub-haplogroups. This reinforces the exceptional genetic status of the dog population in Southern East Asia, and points out Thailand as a possible region from which today's global dog population may have dispersed.

These results are discussed declaratively that the authors of this study identified not only origin of the dogs before 15000 years, but also their concrete origin point for domestication in Thailand.

All these conclusions are based only on 265 recent dogs from Thailand (“(163 new samples and 102 samples from literature”) with comparative analysis with a total 3,254 dogs samples from across the Old World. The last dog data sets include only two studies from 2007 and 2017 years.  

mtDNA data indicate a single origin for dogs south of Yangtze River, less than 16,300 years ago, from numerous wolves. doi:10.1093/molbev/msp195.

and

A cryptic mitochondrial DNA link between North European and West African doi:10.1016/j.jgg.2016.10.008.

These terminal opinion is derived basically only from a few studies cited in the references. A brief analysis for 20 cited papers in ChapterReferences” showed five auto-cited studies, six technical studies (statistical, software, geographic) and only nine other dog studies. Most of the last are cited briefly once in the body of the paper usually in sentences with negative/vice versa position.

As example in section Discussion was cited only five studies [2], [7], [18], [19] and [20].

In sentence

299 An indication of this is that a study of mtDNA in West European pre-Neolithic dogs found exclusively haplotypes belonging to haplogroup C [18], while a study of 16 Neolithic dogs from Southeast Europe showed "normal" frequencies of haplogroup A and B (75% and 20%, respectively) and no haplotype from haplogroup C [19], and our study similarly shows that today's European dogs carry all three major haplogroups, A, B and C, with only 8% carrying haplotype C (Table 1).”

comments of this studies are incorrect and even misleading. European Neolithic dogs (15000 – 6000 Y BP) have A, B, C and D haplogroups not only A, B and C. Haplogroup A is widely distributed in South Europe but also in Nord Europe. Haplogroup D is other important European Neolithic dog group that prevalent in samples from East Europe and Iran.

Finally for me is deeply unclear when and how the Thai dogs arrived in “pre-Neolithic Europe“? What are these human and trade migration routes from SE Asia to Europe and West Africa before Neolithic? Why C and D groups are not identified in studies of ancient SE Asiatic dogs before 2000 Y (doi: 10.1038/s41598-018-27363-8.). Haw authors will comment prevalence of  sub-clade  A2  and  a  cases  of  B  and  A4’5  in ancient SE Asia before 2000Y (doi: 10.1038/s41598-018-27363-8.) but not recent A1, C and D? Why the authors excluded real possibility recent Thai dogs village admixture to be due to long time presence of Europeans in this important trade region on the world?

In my opinion this article must be basically rewritten because it is no need to be commented dogs domestication.

.

Reviewer 2 Report

line 97.- It is necessary to describe the part of materials and methods,DNA extraction, amplification and sequencing, it is very poor. 

line 127.- I think this part "   In conclusion, the mtDNA gene..." is better if you put in the end, because is a conclusion.

Author Response

Point 1: line 97.- It is necessary to describe the part of materials and methods,DNA extraction, amplification and sequencing, it is very poor. 

Response 1: We agree, and have now accordingly given a detailed description of the methods.

Point 2: line 127.- I think this part "   In conclusion, the mtDNA gene..." is better if you put in the end, because is a conclusion.

Response 2: We do not agree that this section is a conclusion, but rather a summary. We have therefore kept it at Line 127, but changed the word "conclusion" to "summary".

Reviewer 3 Report

Zhang et al. sequenced 582 bp of the control region of mtDNA in 163 dogs from Thailand, and combined them with sequences of 102 dogs from Thailand, 3254 dogs from other regions, and 40 wolves previously published, and examined the haplotype diversity. They found the high haplotype diversity with the low proportion of universal haplotypes harboring in many sub-haplogroups in Thailand dogs and suggested Thailand as a possible region of dog origin. The dataset is huge, and the topic is interesting, but there are several issues I found.

Major points

It is true that the control region is highly polymorphic, and many studies in different species have used this region and have been a useful tool to conduct population genetic studies, including aDNA. At the same time, the authors need to note that the fast-evolving control region is unreliable as a molecular clock, and the phylogenetic relationship is not always recapitulated. There is often incongruence of trees between the control region and other regions of mtDNA. The authors did not provide bootstrap values that should be provided, but I suspect the bootstrap supports are very low. In fact, figure 2 clearly shows the inconsistent subclades between the current dataset and the previously published phylogenetic tree based on the mitochondrial genome. In the domestic dogs and wolves including aDNA, quite a few complete mitochondrial genome sequences are published to study the origin of domestic dogs. I recommend to sequence complete mitochondrial genome in Thailand dogs, combine them with the published genome sequences, and analyze if they would like to discuss the origin of the domestic dog, although I do think the authors can rewrite the manuscript using the current dataset more focusing on a different view.

The authors combined the sequences of wolves, however they didn’t show them in the results. Please indicate which haplotypes are derived from wolves in figures 2 and 3. Also, it might be interesting to indicate which haplotypes are Thai Ridgeback dogs data in figures 2 and 3. The authors need to make the figures informative. In addition, there are a bunch of ancient dogs/wolves mtDNA genome sequences published of that control region can be combined and reanalyzed to show the relationships with the Thailand dogs. The authors do argue about wolves and ancient dogs in the discussion, but authors need to show their results clearly.

One of the reasoning the authors suggest the possibility of the Thailand origin of the dogs is that the low proportion of dogs carry UT or UTd in Thailand. For me, this result seems to mean the dog populations in Thailand are fragmented or isolated. The NJ tree in figure 2 also shows the Thailand dogs are not basil (mutations are new). The description in line 102, “The sampled dogs were, except for a few exceptions, collected from remote rural regions with limited influx of foreign dogs,” supports the low proportion of UT or UTd is due to the isolated population.

The authors stated that Thailand dogs showed the highest genetic diversity, but only haplotype diversity was shown. In line 134, it is mentioned that the nucleotide diversity and pairwise genetic distances were also calculated, but I don’t see the data anywhere. Please add the nucleotide diversity in Table 1 because nucleotide diversity can be low even the haplotype diversity is high. The main focus of this manuscript is the high genetic diversity in Thailand dogs, and it is critical to provide this information. Also, haplotype diversity doesn’t seem to be exceptionally high compared to dogs in other regions. The histogram in figure 5 only shows the range from 0.90 to 0.98. Thus it looks as if the Thailand dog has high haplotype diversity; however if the authors add the error bars, which I recommend, I think most population haplotype diversities would overlap.

Figure 3 is too small and complicated, and it is very hard to examine the results. I recommend to make one big MS network, make haplotype circles proportionate to the sample sizes with the different colors for each region (make the pie chart) like Shannon et al. (2015) did. Then move the original figure 3 in the supplementary file.

Minor points

Line 20: Spell out “UTd” and put “UTd” in parentheses here. I had no idea what UTd means when I read the first time.

Lines 184-185: I don’t understand based on what the authors mentioned “in the phylogenetic tree sub-haplogroups a2, a3, a4, a6 and c2 form separate clades.” There are no bootstrap values to support these subclades. Also, as I mentioned in the above, short control region cannot be a reliable molecular clock. The inconsistent subclades between mtDNA genome and high mutating control region is not surprising.

Lines 271-275: It’s not clear whether Thai Ridgeback dogs data is included in figure 5. Were they combined with other Thailand dogs? But as far as I see the Table S1, subregions are not assigned for Thai Ridgeback dogs. Please add the data to figure 5. Also, please indicate Thai Ridgeback dog data in figures 2 and 3.

Line 298: It’s mentioned that Thailand dogs carry all 10 sub-haplogroups, but figure 2 doesn’t support this.

Table 1: Maybe add indents or something to make it clear which ones are sub-regions and which ones are combined regions. For example, the current table has the same indents for Thailand (total in Thailand), four regions in Thailand, and Thai Ridgeback but if you add indents for four regions and Thai Ridgeback, it’ll be easier to see the table. It might be mentioned somewhere in the main text, but also add the caption to explain which locations are combined such as S.E. Asia (I assume Thailand and other regions data are combined for this). By adding the indents, it’ll be clear which regions were included in S.E. Asia as well.

Figure 2: I don’t see “a1.”

Reviewer 4 Report

Thank you very much for your effort in bringing together and analyzing these data. The manuscript is very well written and structured. My few comments are found below. Overall, my strongest requirement is a comparison with the genetic diversity of the domestic cat, as well as a short overview on the situation for the farm animals e.g. sheep, horse, pig.

Lines 68-71: what is the purpose of this information? You can either offer more details or delete the last paragraph.

Line 85: why only 582bp? And the limitations of mtDNA should be discussed also.

Table S1: which are your samples? Please make this clear.

Line 110: could you please provide more details on this software?

Line 147: please change to “as it has already been reported”

South of China and Nigeria also have a high number of haplotypes. What is your hypothesis on this? Lines 194-231 offer some ideas, but I believe this is a situation requiring more arguments.

Line 235: change “markedly” to “significantly”

Line 245-249: any idea on the morphology of these 16 dogs carrying the E haplotype?

Line 248: you mean these dogs are something like hybrids?

Line 251-252. In terms of sampling, you have Thailand very well covered, but data from Europe comes just from 6 countries (according to Fig. 4), which barely represents one quarter of the continent. So this is why, in your dataset, Thailand harbors a superior genetic diversity. Most probably, if you would have an equal sampling effort for Europe or other region in your study, the situation would be different. I think this should be mention in the first paragraph of the discussion.

Line 285: “the sample SIZE is small”

For both supplementary figures, please insert a box with the legend (on the first page would be enough), as in figure 3 in the text. It will make interpretation much easier.

My main concern regards the discussion sector. I am missing a comparison with the genetic situation of cats: where is the cradle of the cat population? Where the dog and the cat domesticated more or less in the same time epoch? And you should add more focus on the fact that early humans, on their nomadic travels, took their pets with them. This would offer a very plausible explanation on why the 18 most common haplotypes/ groups are some widespread. Finally, what about other domesticated animals, like the cow, the horse, the pig. For example, the now ubiquitous chicken is derived from the Red junglefowl, a species of wild bird still found only in Southeast Asia. 

Round 2

Reviewer 1 Report

It appears that the authors of manuscript pretend to do not understand my questions and suggestions and refuse to reinterpret the published papers for Europe and Asia. Practically they refuse to make any changes to the manuscript. In my opinion this manuscript has a main purpose to present Thailand as a dog domestication center and misleads the readers by refusing to acknowledge other published data and including in it misleading tables and figures in the manuscript. I still do not understand how a paper discussion can be made by only 10 references (see my first review). There are many whole paragraphs without cited references.

In conclusion I would like to see correct changes to my suggestions and recommendations to the manuscript.
